# Structural Assembly of Qβ Virion and Its Diverse Forms of Virus-like Particles

**DOI:** 10.3390/v14020225

**Published:** 2022-01-24

**Authors:** Jeng-Yih Chang, Karl V. Gorzelnik, Jirapat Thongchol, Junjie Zhang

**Affiliations:** Center for Phage Technology, Department of Biochemistry and Biophysics, Texas A&M University, College Station, TX 77843, USA; jengyihchang@tamu.edu (J.-Y.C.); karl.gorzelnik@gmail.com (K.V.G.); jirapat.tho@tamu.edu (J.T.)

**Keywords:** single-particle, ssRNA virus, genomic RNA, virus assembly

## Abstract

The coat proteins (CPs) of single-stranded RNA bacteriophages (ssRNA phages) directly assemble around the genomic RNA (gRNA) to form a near-icosahedral capsid with a single maturation protein (Mat) that binds the gRNA and interacts with the retractile pilus during infection of the host. Understanding the assembly of ssRNA phages is essential for their use in biotechnology, such as RNA protection and delivery. Here, we present the complete gRNA model of the ssRNA phage Qβ, revealing that the 3′ untranslated region binds to the Mat and the 4127 nucleotides fold domain-by-domain, and is connected through long-range RNA–RNA interactions, such as kissing loops. Thirty-three operator-like RNA stem-loops are located and primarily interact with the asymmetric A/B CP-dimers, suggesting a pathway for the assembly of the virions. Additionally, we have discovered various forms of the virus-like particles (VLPs), including the canonical *T* = 3 icosahedral, larger *T* = 4 icosahedral, prolate, oblate forms, and a small prolate form elongated along the 3-fold axis. These particles are all produced during a normal infection, as well as when overexpressing the CPs. When overexpressing the shorter RNA fragments encoding only the CPs, we observed an increased percentage of the smaller VLPs, which may be sufficient to encapsidate a shorter RNA.

## 1. Introduction

Single-stranded RNA bacteriophages (ssRNA phages) infect a variety of Gram-negative bacteria [1,2,3,4,5]. The canonical ssRNA phage Qβ has a genomic RNA (gRNA) of 4217 nucleotides, from the 5′ to the 3′ end, encoding the maturation protein (Mat); the major coat protein (CP); the minor capsid protein *A*_1_, which is a *read-through* extension caused by a leaky stop codon for the major CP; and the β-subunit of the RNA-dependent RNA replicase (Rep) [1]. The gRNA of Qβ has been shown to have a defined 3D conformation inside a near-icosahedral capsid whose quasi-symmetry (triangulation number *T* = 3) is disrupted by a single Mat [3,6]. Surprisingly, inside the Qβ capsid, there is a sequestered dimer of CPs interacting with an RNA stem-loop originated from a 5-way junction domain, which also presents a stem-loop to interact with the Mat [6]. Such an RNA domain may form a local nucleation site for the capsid to condense around. However, the RNA sequence of this 5-way junction domain has not yet been identified, though this sequence is essential for linking RNA to the Mat, which binds the bacterial pili [7] and enters the host cell with the viral RNA [8].

Assembly of ssRNA phages is thought to start from the CPs binding to high-affinity stem-loops, either at the “operator”, a stem-loop at the beginning of the replicase gene for each phage, or other “packaging signals” spread throughout the genome [9]. Like many other viruses, the CPs of ssRNA phages can also assemble into virus-like particles (VLPs), which are not infectious [10]. When overexpressing the CPs, many alternative particle sizes and shapes compared to the canonical *T* = 3 icosahedron have been reported, for example, *T* = 1 and *T* = 4 icosahedrons [11,12,13], prolate particles [14,15], and rod-like particles [16]. All of these alternative sizes and shapes of VLPs were observed when the CP was overexpressed from a plasmid, or was otherwise modified. It has not been observed that such a variety of VLPs can be produced from a normal infection by a wild-type ssRNA phage. Increasing the availability of various forms of VLPs is important to encapsidate cargos of different sizes for nanotechnological and biomedical applications, such as for batteries [17,18], vaccines [19,20], and drug delivery [21,22]. A basic understanding of the assembly mechanism of ssRNA phages is also needed to ensure the correct formation of VLPs, and is invaluable for expanding their applications.

In this paper, we first establish a complete model of the gRNA inside of an ssRNA phage, Qβ, using computational structural modeling based on the cryo-electron microscopy (cryo-EM) density map. The model reveals not only the RNA domain that interacts with the Mat, but also all the RNA stem-loops that interact with the CPs for assembly. Such a model is consistent with our subsequent results from the Electrophoretic Mobility Shift Assay (EMSA) and mutagenesis. Additionally, rather than purifying the virus based on infectious activity, we purified the particles by identifying the CPs in different steps, and identified a variety of alternative forms of VLPs produced directly from a wild-type phage infection for the first time. Two forms of these VLPs, namely, the oblate and the D3 symmetrical small-prolate forms, have not been observed before. Taken together, these data show the versatility of the CP of Qβ, and provide a better understanding of viral assembly for ssRNA phages.

## 2. Materials and Methods

### 2.1. Image Processing of the Qβ gRNA

To obtain a better gRNA density, we combined the images of the Qβ particles with and without MurA (the lysis target) bound, collected from the previous study [6] with different pixel sizes. All the images are scaled to a pixel size of 1.25 Å. After the 2D cleaning and the 3D classification using Relion [23], we obtained 86,081 particles with a defined gRNA structure. The unsupervised asymmetric refinement yielded a final map at 6.1-Å resolution before sharpening, and 4.6-Å resolution after sharpening. The gRNA density was less noisy and more complete in the unsharpened map, which was used for the modeling of the RNA.

### 2.2. Modeling of Wild-Type gRNA

To locate the starting point for tracing the entire gRNA structure of Qβ, we first focused on the density attaching to the Mat and the internal CP dimer. This density shows a clear 5-way junction geometry, and is unique in the map of Qβ gRNA [24]. To find the sequence of this 5-way junction, we obtained the local secondary structure from two methods: one is from the phylogenetic prediction [25]; the other is from the minimum-free-energy prediction with sliding-windows of 100–120 nucleotides [26]. Both of these methods show that only the 3′ untranslated region (UTR) can form a 5-way junction with compatible helical lengths for each encompassing stem-loop. Interestingly, the 3′ UTR of a related ssRNA phage, MS2, also interacts with its Mat [2]. More generally, the folds of the 3′ UTRs in all ssRNA phages are very similar, despite the differences in their hosts and gRNA sequences [5]. Therefore, the high-resolution structure of this region from the MS2 was used as a reference to model the structure of the 3′ UTR of the Qβ. Based on the density, the structure of the Mat-bound gRNA stem-loop, U1 of Qβ, is similar to the one of MS2 (Appendix A); therefore, the predicted Qβ U1 could be wrong (Appendix A) [24]. Compared to the predicted secondary structure, one G-C pair in the U1 of Qβ has to be broken to form a bulge with two new G-C pairs at the tip. The modified U1 was refined using the density at 4.4-Å resolution (EMD-8708) [6], and then was incorporated into the entire 5-way junction of the 3′ UTR by Rosetta RNA Denovo [27]. The final model of the 5-way junction fits well with the density after HNMMC [28] and MDFF [29] refinements (Appendix A).

Starting from the structure of the 5-way junction, the entire gRNA structure of the Qβ can be traced using its secondary structure from the phylogenetic prediction [25]. Combining the computational tools of RNA modeling and EM density fitting, including Rosetta RNA Denovo [27], MOSAICS-EM [28], and Situs [30], we first built the atomic models of the Qβ gRNA fragments based on the corresponding segmented densities of the unsharpened map. These segments were then combined to produce a final model of the complete Qβ gRNA, which was then integrated with the capsid and internal coat proteins, and refined using the real-space refinement in PHENIX [31] with the secondary structure restraints.

### 2.3. Purification of Qβ and VLPs

Wild-type Qβ was purified using the CsCl density gradient as before [3]. This technique worked for purifying the large amounts of phage produced from an infection, but when purifying from basal level expression off of a plasmid (pBRT7QB), there was noticeable contamination from large components within the cell, presumably ribosomes. To purify the non-infectious VLPs, either from over-expressing the *cp/A*_1_ or basal-level expression of the U1 and R1 mutants, another purification protocol was established. Single colonies of BL21 (DE3) in the case of *cp/A*_1_ (pET28), or DH10B for wild-type Qβ and the U1 or R1 mutants (pBRT7QB), were used to inoculate starter overnight bacterial cultures with appropriate antibiotics. The starter cultures were used to inoculate 6 × 500 mL cultures with 1:4 aeration grown at 37 °C. The *cp/A*_1_ cultures were induced with 0.5 mM IPTG once the absorbance was at OD600 = ~0.6, then grown for ~4 h. The cells were then spun down, resuspended in Qβ buffer, and lysed using a French Press. The lysate was clarified by spinning at 25,000× *g*, and the supernatant was saved. The supernatant had PEG6000 and NaCl added to 10% *w/v* and 500 mM, respectively. The samples were stored at 4 °C, then spun down at 8000× *g* for 30 min with the supernatant decanted, and the pellet was resuspended in 125 mM NaH_2_PO_4_, pH = 8.0. To remove excess PEG from the sample, the resuspended pellet was mixed with an equal volume of chloroform, and vortexed. The suspension was spun down at 40,000× *g*, and the aqueous layer was removed and extracted with chloroform, in the same manner, four more times. After the PEG was extracted with chloroform in this manner, the samples were concentrated using a 10 kDa MWCO centrifugal concentrator until the volume was ~3–4 mL. The samples were loaded onto a HiPrep 26/60 Sephacryl S500 column, run with 125 mM NaH_2_PO_4_. Between different samples, the column was washed with 0.5 M NaOH and then buffer. Purifying wild-type and the U1/R1 mutants were the same as purifying the *cp/A*_1_ VLPs, except that they were not overexpressed. Instead, the phage/mutants were produced from basal-level transcription from pBRT7QB. Cells were removed by spinning down at 4000× *g*, then the supernatant was filtered and PEG-precipitated as above.

### 2.4. Western Blots

The 3′ mutants, plus wild-type, were produced from the plasmid in *E. coli* DH10B (F-). Cultures of 5 mL were inoculated from a single colony, and grown with 100 µg/mL ampicillin. The cultures were grown overnight, then diluted to OD600 = 0.5, with the OD measured again after dilution. The samples were spun down, then the supernatant was incubated on ice for 30 min before the TCA precipitation. Samples were TCA precipitated by adding ice-cold TCA to 10% final volume, and acetone to 50% final volume. The sample was spun down at 18,000× *g* for 15 min at 4 °C, and the supernatant aspirated off. The pellet was washed twice with ice-cold acetone, then resuspended in a sample loading buffer, boiled for 15 min, then loaded onto a 4–20% Tris-Tricine gel. The gels were run for ~1 h, then transferred to a membrane using a semi-dry apparatus, and blocked for at least half an hour with 2% milk in TBS with 0.1% Tween 20. The buffer was replaced with a fresh buffer which had an α-A_2_, derived against a synthetic peptide PKLPRGLRFGA (from the N-terminal end of A_2_) [32]. The antibody was used at a 1:1000 dilution. After applying the antibody overnight, the membranes were washed five times for at least five minutes, each time with TBST. The sample was then incubated with a secondary antibody (1:3000 dilution of goat-anti-rabbit).

### 2.5. Electrophoretic Mobility Shift Assay

The RNA fragments were in-vitro transcribed using a HiScribe™ T7 High Yield RNA Synthesis Kit (New England Biolabs, Ipswich, MA, USA). The linear DNA templates for in-vitro transcription were amplified by PCR from the Qβ cDNA. The binding reaction contains 1 μM of the RNA fragment, 10 µM of purified MBP tagged Mat (MBP-A_2_) or 10 µM of purified MBP (6xHis-MBP), 2 U/µL of SUPERase•In™ RNase Inhibitor (Thermo Fisher Scientific, Houston, TX, USA), and the binding buffer (20 mM Tris-HCl pH 7.6, 150 mM NaCl). The reactions were incubated for 20 min at 37 °C. After the incubation, the 1x EMSA loading dye was added to the reaction. The final volume of 10 µL of the reaction was loaded into 5% Native-TBE polyacrylamide gel, and separated at 200 mV for 30 min at 4 °C. The gel was then stained with SYBR™ Green and subsequently stained with SYPRO™ Ruby using the Electrophoretic Mobility Shift Assay (EMSA) Kit (Thermo Fisher Scientific, Houston, TX, USA).

### 2.6. Cryo-EM Specimen Preparation and Data Collection

The cryo-EM grids were prepared by applying 3 µL of the sample solution to a C-flat 1.2/1.3 400-mesh Holey Carbon Grid (Protochips, Morrisville, NC, USA) at 22 °C with 100% relative humidity, and vitrified using a Vitrobot Mark III (FEI Company, Hillsboro, OR, USA). Movie stacks of the gel-filtration purified Qβ particles (wild-type and the *cp/A*_1_ overexpression) were collected on an FEI Tecnai F20 cryo-electron microscope (TF20, FEI Company, Hillsboro, OR, USA) operated at 200 kV. Data were recorded using SerialEM [33] on a K2 Summit direct detection camera (Gatan, Pleasanton, CA, USA) at a nominal magnification of 29,000×, yielding a pixel size of 1.25 Å. Each movie stack has 33 frames with 0.2 s per frame for a total exposure time of 6.6 s, yielding a total dose of ~36 e^−^/Å^2^.

### 2.7. Cryo-EM Data Processing

The movie stacks of the gel-filtration purified particles imaged from TF20 were aligned and filtered by MotionCor2 [34] (1.25 Å/pixel). The images of the CsCl-purified particles are from the previous collection using JEM3200FSC [6] (2370 micrographs, 1.216 Å/pixel). All the Qβ particles were picked by Gautomatch (K. Zhang, MRC LMB (https://www2.mrc-lmb.cam.ac.uk/research/locally-developed-software/zhang-software/), accessed on 22 December 2020), and then downscaled by two times. The particles were separated by their 2D classified images (forms of *T* = 4, oblate, *T* = 3, prolate, and small prolate, respectively). In total, for the CsCl purification, we have 1138 *T* = 4 particles, 4316 prolate particles, and 1489 oblate particles; for gel-filtration purification, combining wild-type Qβ and over-expressed *cp*/*A*_1_, we have 1635 prolate particles, 38,125 oblate particles, and 6842 small prolate particles. Due to the small numbers of particles of each form in different collections, for the *T* = 4 and prolate VLPs, we integrated the particles from the TF20 collection to the data from the JEM3200FSC collection, and rescaled all the particles to the pixel size of 2.432 Å, which is 2-times downscaled compared to the original pixel size of the JEM3200FSC collection. Similarly, the merged particles of the oblate and small prolate VLPs were rescaled to 2.5 Å, which is 2-times downscaled compared to the original pixel size of the TF20 collection. During the refinements, we applied icosahedral symmetry for *T* = 4 particles, D5 symmetry for prolate and oblate particles, and D3 symmetry for small prolate particles. The refinements yielded the maps of *T* = 4, prolate, oblate, and small prolate particles at 5.6-Å, 6.1-Å, 6.2-Å, and 8.9-Å resolutions, respectively (Appendix A). The refinements of the particles without downscaling did not improve the final resolutions of the maps.

### 2.8. Modeling of VLP Capsids

To build the atomic models for the Qβ capsids of different forms, the structure of the CP from an icosahedral refinement (PDB ID: 5KIP) was first rigidly fit and subsequently refined into each VLP capsid. The refinements were performed by using RosettaCM [35] with icosahedral symmetry for the *T* = 4 particle, D5 symmetry for the prolate and oblate particles, and D3 symmetry for the small prolate particle.

## 3. Results

### 3.1. The Complete gRNA Model inside the Qβ Virion

From our cryo-EM density map, we have built a complete atomic model of the Qβ virion, including the entire 4217-nucleotide long gRNA in its dominant conformation in the capsid (Figure 1, Video S1). The gRNA folds domain by domain (Video S2) with the 3′ UTR (the red-ribbon portion of the model in Figure 1) packaged close to the Mat, and the 5′ UTR (the blue-ribbon in Figure 1) far away from the Mat. This orientation of the gRNA packaging is consistent with the EMSA result, in which the purified Mat has a much stronger affinity towards the 3′ end of the gRNA compared to the 5′ end (Appendix A). The result that, in Qβ, only the 3′ end of the gRNA binds to the Mat, is in agreement with the recent observation in another ssRNA phage MS2 [2], but in contrast to the previous results from in vitro RNase protection assays, which showed the Mat of MS2 binds to both the 5′ and 3′ regions of the gRNA [36]. The RNase protection assay result for MS2 can be interpreted as the Mat binding RNA in vitro, whereas in the infectious virus, it is always bound to the 3′ end, as seen by the cryo-EM structure.

The RNA 5-way junction domain for Qβ, which interacts with the Mat, the internal CP dimer, and one exposed CP dimer, is determined to be the 3′ UTR of the gRNA (Figure 1D and Appendix A). The last stem-loop in the entire gRNA, U1, specifically interacts with the Mat, and the stem-loop, U2, interacts with one exposed CP dimer (Appendix A). Unique to the Qβ is that the last stem-loop of the replicase gene, R1, specifically interacts with the internal CP dimer. The last six nucleotides of the entire gRNA, including the CCA tail, fold back to be part of the stem-loop R1 by pairing with residues 4124 to 4129 (Appendix A) [24]. This compact RNA 5-way junction domain has a biotechnological implication that serves as an adapter to the Mat for an RNA of interest to be delivered into a target host. Given the fact that most ssRNA phages have a conserved motif for the 3′ UTR of the gRNA [5], such a linkage between the 3′ UTR and the Mat may be conserved in ssRNA phages, leading to a common mechanism for gRNA packaging and delivery. The specific RNA structures within the 5-way junction domain are required for Qβ infectivity, as mutations within either the U1 or R1 stem-loops do not affect the production of phage-like particles, but abolish infectivity (Appendix A), primarily due to their requirement for the incorporation of the Mat into the capsid. Consistently, Western blots of the Mat show that, for these mutants, the Mat failed to be incorporated into the phage capsids (Appendix A).

One striking feature of the gRNA fold in the Qβ is a ~200-Å long helical structure that horizontally spans the equator of the capsid when the Mat is defined as the north pole (Figure 1C) [3]. In the complete model of the gRNA, it was determined that this long helical structure is in fact formed by shorter RNA stem-loops connected through two kissing loops (indicated by orange and red boxes in Figure 1C, zoomed-in Figure 1E). One kissing loop is formed between residues “2844-UCGU-2847” of stem-loop R22 in the gene of the replicase and residues “1876-ACGA-1879” of stem-loop RT4 in the gene of the *read-through* protein *A*_1_ [37]. The other is a branched kissing loop [38] formed between residues “2811-UUUUC-2815” of stem-loop R23 and residues “2749-GAAAA-2753” of bulge R24 in the gene of the replicase. Sequence alignment shows the nucleotides forming these two kissing loops are conserved in Qβ-like phages (Appendix A). In addition, bulge R24 encodes amino acids Arg133 and Lys134 in the β-subunit of Qβ replicase. These two amino acids are important for Qβ replicase to recognize the gRNA template during the initiation of the replication [39]. The sequence conservation for these amino acids has been coupled into the structural morphology of the viral particles, as mutations in bulge R24 may disrupt the formation of the kissing-loop, leading to instability of the gRNA fold, which may cause an incorrect RNA packaging. Therefore, this long-distance interaction may have imposed selective pressure against mutations in this region.

### 3.2. Operator-like RNA Stem-Loops Support the Formation of CP Pentamers

The capsid for an infectious ssRNA phage virion has a near icosahedral *T* = 3 capsid, which consists of twelve pentamers and twenty hexamers of the CPs (Figure 2A). One CP dimer, at the two-fold axis on the shared edge of two hexamers, is replaced by the Mat, resulting in a total of 89 CP dimers on the capsid. For Qβ, an extra internal CP dimer interacts with stem-loop R1 of gRNA inside the capsid, still making a total of 90 CP dimers. Such an icosahedron can be unwrapped onto a hexagonal lattice with each thick edge representing a CP dimer (Figure 2B). Out of the 77 stem-loops within the Qβ gRNA, 59 have a distance less than 5 Å from the protein components, which are the CP shell, the Mat, and the internal CP dimer. Of these stem-loops, 33 interact with CP dimers in a fashion similar to stem-loop 34, the actual translational operator, wherein the backbones of the RNA stem-loops follow a similar trace relative to the CP-dimer (Figure 2C, Appendix A), although with diverse sequence motifs (Appendix A). These operator-like stem-loops are indicated as circles on the hexagonal lattice (Figure 2B). The locations of these operator-like stem-loops are throughout the entire gRNA sequence (labeled in bold black font in Figure 2D), and are proposed to be directly involved in the assembly of the viral capsid [40]. Interestingly, some of the stem-loops interact with the CP dimers in a different fashion from the operator-like stem-loops (with the backbones of these stem-loops folding in an alternative manner), and are classified as two types. One type of the stem-loops points toward the center of the CP dimers (termed CP-sandwiched stem-loops, Appendix A), and the other type is anchored on just one of the CPs in a CP dimer (referred to as CP-anchored stem-loops, Appendix A). The CP-sandwiched stem-loops may have relatively strong affinities to the CPs, and recruit CP dimers to facilitate the capsid assembly, whereas the CP-anchored stem-loops may have lower affinities to the CPs, but can still contribute to the stability of the viral capsid (Appendix A). The locations of these CP-sandwiched and CP-anchored stem-loops in the entire gRNA sequence are labeled in normal black and grey, respectively, in Figure 2D.

In the icosahedral structure of ssRNA phages, the CP dimers can be classified as A/B dimers (located on the edge between a pentamer and a hexamer) or C/C dimers (located on the edge between two hexamers) [41,42]. Notably, for the 33 operator-like stem-loops in Qβ, 24 of them interact with A/B dimers; 8 of them interact with C/C dimers; and 1 interacts with the internal CP dimer. Surprisingly, in Qβ, the actual translational operator, stem-loop 34, interacts with a C/C dimer on the capsid (labeled as 34 in Figure 2B). In MS2, it was reported that binding of the 19-nt long RNA fragment containing the translational operator will change the C/C dimer into A/B dimer in solution [43]. However, in the context of the entire gRNA, such a conformational change induced by the operator-binding may also be constrained by the fold of the gRNA and the location of this stem-loop in the gRNA, as well as the packing of the CPs within the capsid shell. Still, the operator-like stem-loops preferentially bind to the pentamers on the capsid, with eleven out of the twelve pentamers having at least one operator-like stem-loop bound. Strikingly, Pentamers I, II, and III have at least four constituting A/B dimers bound with an operator-like stem-loop (Figure 2E). That is almost half of the operator-like stem-loops within the whole gRNA clustered inside this local patch of the capsid. These stem-loops are mostly located within the 5′ half of the gRNA. As the gRNA is synthesized from the 5′ end to the 3′ end, in the event of co-replicational genome packaging [44], it is intuitive that a local patch of the capsid would form first on the 5′ half of the gRNA. Such an immediate packing of the nascent RNA by the CPs may also prevent the replicated RNA from annealing back to the template strand of negative-sense RNA.

### 3.3. Qβ VLPs of Different Forms Are Made along with Infectious Virions

Recently, *T* = 4 capsids have been observed when overexpressing the covalently-linked CP dimers in ssRNA phages PP7 and MS2 [12,13]. Here, we have observed under normal wild-type phage infection, *T* = 4 capsids are also made, albeit a small percentage (0.5%) of the overall number of particles (Figure 3A and Appendix A, and Appendix A). Surprisingly, we have also observed that about 1.7% of the VLPs show a “walnut-like” prolate shape (Appendix A). A 3D reconstruction of these particles reveals that they are a prolate form with *T* = 3 and *Q* = 4, where *T* and *Q* describe the triangulation number of the two caps and the elongated faces, respectively [45]. More interestingly, we also observed smaller particles compared to the *T* = 3 VLPs. To better analyze the structures of these smaller particles, we modified our phage purification protocol by using gel-filtration chromatography instead of the density gradient (see Materials and Methods, Appendix A). Although the number of *T* = 4 or prolate particles does not increase, to our surprise, almost a quarter of the VLPs purified by the modified protocol are smaller than *T = 3* particles. Further structural characterization of these smaller particles revealed two previously unreported forms of VLPs. One is the oblate form with *T* = 3 and *Q* = 2; the other is a small prolate form elongated along the 3-fold axis from the two *T* = 1 caps. The elongated faces of the small prolate consist of two triangulation numbers, *Q*_1_ = 2 and *Q*_2_ = 3 (Appendix A). Figure 3A shows the five forms of the VLPs from the wild-type Qβ, each containing 240, 210, 180, 150, and 132 CP copies for the *T* = 4, prolate, *T* = 3, oblate, and small prolate forms, respectively (Figure 3B). Though all of the 5 forms contain 12 copies of the pentamers, they consist of 30, 25, 20, 15, and 12 copies of the hexamers for the *T* = 4, prolate, *T* = 3, oblate, and small prolate forms, respectively. When produced from an infection event, and purified via size-exclusion chromatography, the population of the oblate and the small prolate forms are around 21% and 3%, respectively. In the density-gradient-based purification methods, these smaller particles may have been left out or ignored due to their different density compared to the regular *T* = 3 particles, and the fact that they are non-infectious.

When over-expressing the gene containing only the *cp* and *read-through* (referred to as *A*_1_) fraction of the Qβ genome from a plasmid, the percentage of the oblate and small prolate forms increase further to 35% and 12%, respectively (Figure 3C). The increase of these smaller particles may be due to the presence of the shorter RNA (expressing just the 987nt for the *cp* and *A*_1_ as opposed to the entire 4217nt in the gRNA). There are also host RNAs with operator-like sequences which may also be encapsidated by smaller particles of the oblate and small prolate forms [46]. Additional densities inside the capsid were observed when performing asymmetric reconstructions of classified particles of the oblate and small prolate forms; however, due to the lower resolution and potential heterogeneity of the RNA, it is not feasible to model the RNA inside these VLPs (Appendix A).

### 3.4. The Plasticity in the Hexamers Supports the Formation of Different VLPs

All these different forms of the VLPs confer different curvatures in the capsids. To investigate the underlying structures that support these different morphologies, given that they are formed by the same CP, we analyzed the assembly and structural plasticity in the CPs. Notably, all the CP dimers for these different forms of the VLPs exhibit small structural variations, which are mostly localized in the flexible loop regions (Appendix A). The same holds true for all of the pentamers in every form (Appendix A). Surprisingly, large structural variations exist within the hexamers in these different forms. Based on the relative locations of hexamers in the capsid and the symmetry of the VLP, the hexamers can be classified into different types (Figure 4A) based on three arch angles, θ, ϕ, and Ψ, describing three opposite CP subunit pairs, respectively (Figure 4B). For the *T* = 3 and *T* = 4 VLPs, both are formed by only one type of hexamer. For the oblate and small prolate VLPs, there are two types of hexamers in each form. For the prolate VLP, there are three types of hexamers. This would give rise to a total of nine types of hexamers (colored differently in Figure 4A,C). Though the ɸ and Ψ angles fluctuate between 150° and 160° for different forms of the VLPs, the θ angle can change dramatically from 130° to 180°. Therefore, we classify the nine hexamers into four different groups based on the values of θ Ψ (Figure 4D–G). The “normal arched” type, which comes from the prolate, oblate, small prolate, and regular *T* = 3 capsids, has a θ angle of ~155°. The “over arched” type, which provides more curvature with a θ angle of ~130°, includes hexamers from the oblate and small prolate VLPs. The *T* = 4 capsid has one type of hexamer with a θ angle of ~170°, defined as a “less arched” type. Interestingly, the hexamer located on the equator of the prolate capsid has an arch angle of ~180°, which is defined as a “straight” type. Such large plasticity within the hexamers of the Qβ CPs enables the assembly of different forms of the capsids (Video S3). Previously, it has been reported that some Qβ CP mutants can form rod-like particles [16], which should be elongated using tubes of the straight-type hexamers.

To further investigate how plasticity within the hexamers is allowed, we examined the interface between two neighboring CP monomers within one hexamer. This interface is formed between the FG-loops of neighboring subunits, and between the C-terminus of one CP and the CD/DE-loops of its neighbor (Appendix A). Though the neighboring FG-loops are linked through disulfide bonds [42], which are always maintained among all these different types of the hexamers, the interactions drastically change between the C-terminus of one CP and the CD/DE-loops of its neighbor (Figure 4H). If we align the CP a in a hexamer (the top CP in the hexamer in Figure 4B) from the over arched type to the straight type of the hexamer, the CD-loop and DE-loop of CP f move 14 Å away from and 6 Å towards the C-terminus of CP a, respectively. Such a relative rocking of the CD/DE-loops around the C-terminus of the neighbor allows subunits to tilt up or curve down with each other to achieve different curvatures of the hexamers.

## 4. Discussion

In this paper, we have presented the complete model of the Qβ gRNA in its capsid, which provides insight into the assembly process of the wild-type Qβ in vivo. The location of the operator-like RNA stem-loops in the entire gRNA suggests an order for the RNA packaging and capsid formation in Qβ (Figure 5A). In packaging a nascent gRNA that is still coming out of the replicase, the first few operator-like stem-loops at the 5′ region of the gRNA, which are synthesized at the beginning, bind CP dimers to coordinate the formation of Pentamer I. Other pentamers and hexamers are then formed, with the last stem-loop in the gRNA, U1, binding to the Mat to be incorporated into the capsid. Intuitively, such a gradual folding and packaging of the gRNA may confine the flexibility of the gRNA, and help a ~4000-nt long RNA fold into a defined 3D structure that fits into a small capsid. The proposed model for RNA-mediated capsid assembly is consistent with our previous observation that the capsid of the wild-type Qβ deviates slightly from an ideal icosahedral cage [3,6], particularly by exhibiting a crack in the capsid around Pentamer XI (Figure 2A), where the β-region of the Mat points. Such an imperfection of the capsid suggests this patch of the CP shell is formed at the end of the capsid assembly when gRNA is folded, and packaged with very little room for the adjustment of the capsid to completely close up.

Through a native infection without engineering the amino acid sequences, we found that Qβ CP can form diverse VLP capsids, including the *T* = 3 icosahedral, *T* = 4 icosahedral, prolate, oblate, and a small prolate form elongated along the 3-fold axis. Though there are always twelve copies of the pentamers in different forms of the Qβ VLPs, the numbers of the hexamers vary significantly. Interestingly, the geometries of a few forms of these capsids are related. For example, one can convert an oblate capsid to a *T* = 3 capsid by separating the two end-shells from the equator and adding five more hexamers where the two half-shells rejoin (Figure 5B). The same procedure can be repeated to generate a prolate capsid from a *T* = 3 capsid. One can even repeatedly add hexamers in a multiplication of five in the middle of the prolate to obtain the rod-like VLP particles previously engineered [16]. Besides the 5-fold non-icosahedral capsids, Qβ can also form a small prolate VLP elongated along the 3-fold axis, which is a new conformation of the virus capsid. In the elongated face, different from the 5-fold capsids having the same triangulation number, the 3-fold small prolate consists of two different triangulation numbers, *Q*_1_ = 2 and *Q*_2_ = 3 (Appendix A). Moreover, the triangulation number of the two caps in the small prolate is *T* = 1, which means that Qβ has the potential to form a *T* = 1 icosahedral VLP. However, we did not find the *T* = 1 capsids in this study.

It has not been studied how the CPs structurally protect the RNA of different lengths. Our results, showing that over-expressing the 987-nt *cp/A*_1_ gene leads to more percentages of the oblate (from 21% to 35%) and the small prolate (from 3% to 12%) forms, suggest the length of the RNA being encapsidated may affect the ratio of different forms of the VLPs. In the wild-type Qβ, the entire wild-type gRNA forms operator-like stem-loops (Figure 1 and Figure 2) to coordinate the pentamers for the assembly, as well as the long helical structure joined by kissing-loops (Figure 1E) that set the diameter of the capsid. The mechanism by which VLPs are assembled around shorter or heterologous RNAs is still not clear. However, the diversity of the forms of VLPs may allow manipulations of the ssRNA phage CPs to package RNA of different lengths for various biomedical or biotechnological applications. For example, multiple RNA vaccines have been recently approved to prevent SARS-CoV-2 infections. The use of VLPs may be a way to reduce the cold storage demands for these vaccines [20].

## Figures and Tables

**Figure 1 viruses-14-00225-f001:**
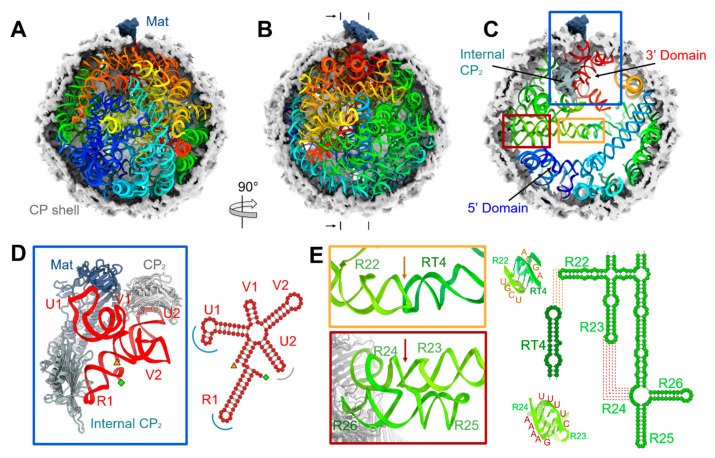
The complete model of the dominant gRNA fold in Qβ. (**A**) The cut-open view of the Qβ virion to display the gRNA, shown as ribbons and rainbow-colored from the 5′ end (blue) to the 3′ end (red). The CP shell and Mat are shown as electron density, colored light grey, and dark blue, respectively. (**B**) The view from Panel A vertically rotated 90° with half of the CP shell removed. (**C**) Cross-section from between the marks in Panel B and viewed from the black arrows. The internal CP dimer (Internal CP2), and the 5′ and 3′ gRNA domains are light blue, blue, and red, respectively. (**D**) Left: zoomed-in view within the blue box in Panel C showing the gRNA 5-way junction domain (red) interacting with the Mat (dark blue), an exposed CP dimer (CP2, grey), and the internal CP dimer (Internal CP2, light blue). The 5′ and 3′ ends of the 5-way junction domain are labeled as green diamond and orange triangle, respectively. Right: secondary structure of the gRNA 5-way junction domain. The dark blue, grey, and light blue arcs denote the interactions with the Mat, and the exposed and internal CP dimers, respectively. (**E**) Left: zoomed-in views within the orange and red boxes in Panel C showing the two kissing loops that link helices R22, RT4, R23, and the bulge R24. The orange and red arrows indicate the locations of the kissing loops. Right: Models of the two kissing loops and their secondary structures showing the base pairing within the sequence.

**Figure 2 viruses-14-00225-f002:**
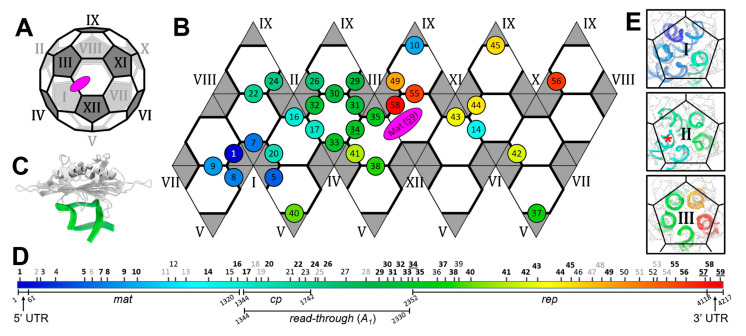
Operator-like RNA stem-loops in the gRNA that interact with the capsid shell. (**A**) The near icosahedral capsid of the Qβ is illustrated as a cage with the Mat labeled as a magenta oval. The twelve pentamers are dark grey, and labeled from I to XII. (**B**) The cage is unwrapped to display the capsid as a 2D map, with each thick line representing a CP dimer. The asymmetric A/B dimer is between a grey pentamer and a white hexamer. The symmetric C/C dimer is between two white hexamers. The rainbow-colored circles label the operator-like stem-loops from the 5′ (blue) to 3′ (red) ends of the gRNA. (**C**) A model showing an operator-like RNA stem-loop interacting with a CP dimer. (**D**) The sequence of the Qβ gRNA is rainbow-colored. The UTRs, the genes of maturation protein (mat), coat protein (cp), replicase (rep), and *A*_1_ protein (*read-through* or *A*_1_) are labeled. The 59 stem-loops, which have a distance of less than 5 Å to the capsid proteins, are numbered and labeled on the sequence with operator-like (bold black), CP-sandwiched (black), and CP-anchored (grey) stem-loops labeled. The actual translational operator and the stem-loop that interacts with the internal CP dimer are labeled by the underlined numbers 34 and 57, respectively. The underlined 59 indicates the stem-loop which interacts with the Mat. (**E**) The operator-like RNA stem-loops (rainbow-colored) interact with the CPs in Pentamers I, II, and III. The red star labels one CP-sandwiched RNA stem-loop (number 15 in Pentamer II).

**Figure 3 viruses-14-00225-f003:**
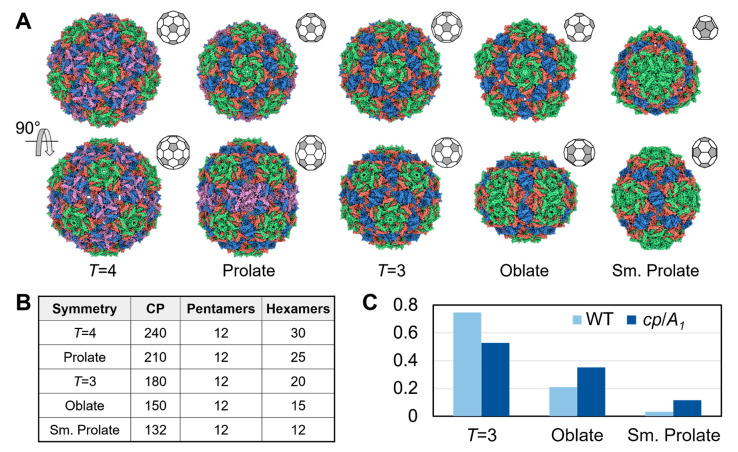
Structures of the Qβ VLPs. (**A**) Top: cryo-EM structures of the Qβ VLPs of *T* = 4, prolate, *T* = 3, oblate, and small prolate symmetries from left to right. The corresponding symmetry cages are drawn with pentamers and hexamers colored grey and white, respectively. Bottom: the same structure horizontally rotated 90°. (**B**) The table shows the number of CPs, pentamers, and hexamers for each form of the VLPs. (**C**) The population for the *T* = 3, oblate, and small prolate VLPs produced by the wild-type Qβ infection (light blue) and over-expressing the *cp/A*_1_ gene (deep blue). Samples were purified by gel-filtration chromatography.

**Figure 4 viruses-14-00225-f004:**
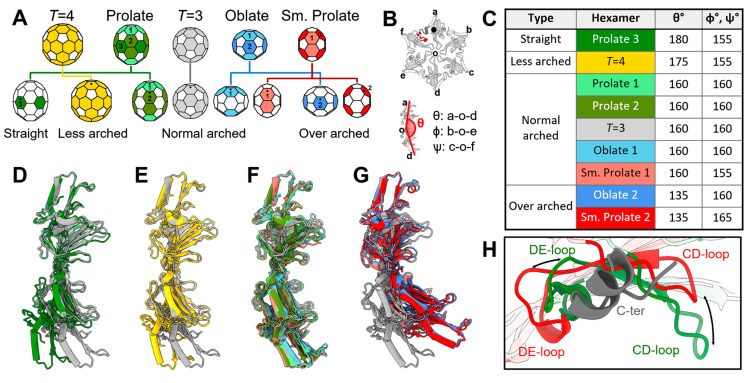
Plasticity in the hexamers of different Qβ VLPs. (**A**) Different forms of the VLPs with the pentamers in white and hexamers colored. The conformations of the hexamers are classified to be “straight”, “less arched”, “normal arched”, and “over arched”. (**B**) The model of a hexamer with the six subunits labeled from a to f. The black dot indicates the relative orientation of the hexamer in Panel A. The curvature of the hexamer can be defined by three arch angles, θ (between subunits a and d), ϕ (between subunits b and e), and Ψ (between subunits c and f). (**C**–**F**) Structural overlay of the hexamers in each type (colored as in Panel A) against the regular *T* = 3 hexamer (grey). The models are aligned based on subunit a of the hexamers as labeled by a black dot in Panels A and B. (**G**) The table showing the values of θ, ϕ, and Ψ for each form of the VLPs. ϕ and Ψ have the same value for each VLP form due to the symmetry. The names of the hexamers are highlighted with the corresponding color in Panel A. (**H**) The relative motion between the C-terminus (dark grey) of one subunit and the CD/DE-loops of its neighbor. The black arrows indicate the movements of the CD/DE-loops between the over arched (red) and straight (green) hexamer conformations. The location of the C-terminus and the CD/DE-loops are labeled in Panel B with the corresponding colors, grey and red, respectively.

**Figure 5 viruses-14-00225-f005:**
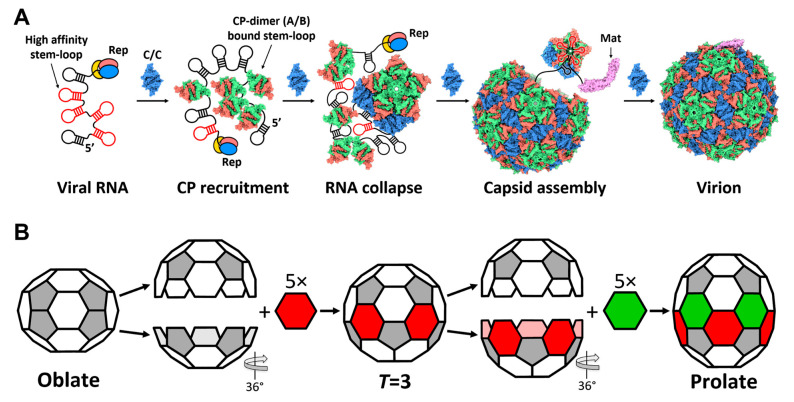
Proposed model for the co-replicational assembly of the Qβ virion, and the geometrical relationship between the oblate, *T* = 3, and prolate VLPs. (**A**) Proposed model for assembly of the Qβ virion along with the gRNA replication. The gRNA synthesized by the replicase (Rep) presents several high-affinity stem-loops for CP dimers, which, upon binding to CP dimers, change their conformations from C/C to A/B. The protein–protein interactions of CP dimers on the growing capsid facilitate RNA collapse, and form the first pentamer, then other pentamers and hexamers. Almost towards the end of the assembly (after the last stem-loop, U1, has been synthesized), the Mat binds to the 3′ UTR, and is incorporated into the capsid to form a complete virion. (**B**) The geometrical relationship between the oblate, *T* = 3, and prolate VLPs. The conversion from the oblate to *T* = 3 VLPs can be achieved through the division of the oblate VLP, perpendicular to the 5-fold axis, into two hemispheres, with one hemisphere rotated 36° relative to the other, followed by the insertion of five hexamers (red). The elongation from the *T* = 3 to prolate VLP follows the same protocol with an addition of five more hexamers (green).

## Data Availability

The cryo-EM maps and models of the Qβ VLPs for the forms with *T* = 4, prolate, oblate, and small prolate symmetries are deposited in the EMData Bank with accession IDs EMD-23321, EMD-23322, EMD-23323, and EMD-23324; and Protein Data Bank (PDB) with accession IDs, 7LGE, 7LGF, 7LGG, and 7LGH, respectively. The atomic model of the complete Qβ virion is deposited in PDB with accession ID 7LHD, and the related cryo-EM map can be accessed by EMD-23336, including an unsharpened map in the same entry.

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
