# Peer review of "Structural Assembly of Qβ Virion and Its Diverse Forms of Virus-like Particles"

_viruses, 2022, doi:10.3390/v14020225_

Round 1

Reviewer 1 Report

I found this to be an important contribution to the field of ssRNA phage structure and to virus structure and assembly generally. The details of genome RNA structure are fascinating. The nature of maturase-RNA interactions is revealed, as are the numerous interactions of stem-loops with coat protein dimers. I found myself wondering how the various operator-like stem loops compare to the canonical operator sequence and secondary structure, i.e. how well do they conform to the known RNA structural requirements for coat protein interaction? This is perhaps something to consider for inclusion in the supplementary information.

The description of diverse alternatives to the usual T=3 structure was also interesting. The apparent plasticity of Qß structure has implications for virus assembly generally, as well as for potential structural manipulation of virus-like particles for practical purposes.

Author Response

We thank the reviewer for the comments. We have now added a Panel B of Supplement Table 1, in which we now show the sequence motifs and secondary structures of the operator-like RNA stem-loops. Interestingly, these operator-like RNA stem-loops have more diverse sequence/secondary structure motifs as compared to the actual operator. We have added a sentence on Line 4 of Page 7 of the revised manuscript to indicate this. 

Reviewer 2 Report

The paper by Chang et al uses the cryo-EM technique to gain insights into the mechanism of Qβ virion assembly. The manuscript is very well-written and great care has been taken to guide the readers through the structural data. There are no major concerns, as a minor point, a more in-depth discussion about the importance of this study in the context of biotechnology applications will be useful.

Author Response

We thank the reviewer for the comments. Our results reveal the detailed RNA-capsid interactions for an ssRNA phage and a diverse form of virus-like particles that may be used for packaging RNAs of different sizes. As for the biomedical or biotechnological applications, this may suggest an alternative strategy for RNA vaccine development and delivery. "For example, multiple RNA vaccines have been recently approved to prevent SARS-CoV-2 infections. The use of VLPs may be a way to reduce the cold storage demands for these vaccines [20]. " We have added the above discussion in the parentheses at the end of the second paragraph on Page 12 of the revised manuscript.